# Peptide binder design with inverse folding and protein structure prediction

Patrick Bryant [1,2] & Arne Elofsson [1,2 ✉]

The computational design of peptide binders towards a specific protein interface can aid diagnostic and therapeutic efforts. Here, we design peptide binders by combining the known structural space searched with Foldseek, the protein design method ESM-IF1, and AlphaFold2 (AF) in a joint framework. Foldseek generates backbone seeds for a modified version of ESM-IF1 adapted to protein complexes. The resulting sequences are evaluated with AF using an MSA representation for the receptor structure and a single sequence for the binder. We show that AF can accurately evaluate protein binders and that our bind score can select these (ROC AUC = 0.96 for the heterodimeric case). We find that designs created from seeds with more contacts per residue are more successful and tend to be short. There is a relationship between the sequence recovery in interface positions and the pIDDT of the designs, where designs with ≥80% recovery have an average pIDDT of 84 compared to 55 at 0%. Designed sequences have 60% higher median pIDDT values towards intended receptors than non-intended ones. Successful binders (predicted interface RMSD ≤ 2 Å) are designed towards 185 (6.5%) heteromeric and 42 (3.6%) homomeric protein interfaces with ESM-IF1 compared with 18 (1.5%) using ProteinMPNN from 100 samples.

[1] Science for Life Laboratory, 172 21 Solna, Sweden. [2] Department of Biochemistry and Biophysics, Stockholm University, 106 91 Stockholm, Sweden.
✉email: arne@bioinfo.se

Designing peptide binders towards specified protein interfaces is a highly coveted goal with major impacts on pharmaceutical development[1]. Over the past six decades, there has been an 8% average annual increase in approved peptide drugs, and global sales exceed $50 billion yearly[2]. Previously, the state-of-the-art for computer-aided binder design[3] relied on docking scoring programs which are now significantly outperformed by AlphaFold2 (AF)[4–6]. The resulting outcome of these classical methods is that 1/100,000 designed sequences bind to their target structures[3]. Alternatively, experimental techniques such as directed evolution can be used together with classical machine learning[7,8] to improve the outcome coupled with phage display[9]. However, none of the experimental methods can be used to design a binder towards a certain interface region, potentially resulting in binding optimisation towards unintended protein regions.

Recent developments in deep learning, mainly based on AF, have shown great promise for structure prediction of protein-protein[4,5] and protein-peptide[10] interactions, as well as protein design[11,12]. A recent protein design method, ProteinMPNN[13], further improved these achievements and created proteins with significantly higher solubility than AF alone.

Protein design methods are evaluated in terms of overall sequence recovery, which is arguably a bad measure if one wants to design binders. The informative metric of binder sequences is the recovery of interacting residues. A shift of only one residue may result in 0% sequence recovery, but the interacting residues may stay intact. The only possibility to evaluate this aspect, short of obtaining experimental structures, is through structure prediction.

Reevaluating previous designs[3] using AF as a scoring function reports experimental success rates of close to 90% in some cases compared to only 0–5% using physics-based calculations[14]. These methods have been applied to design binders with and without scaffolds and known binding motifs[15]. Other methods, such as MaSIF, utilise learned interaction potentials and protein scaffold search to design binders[16].

By inverting the structure prediction process, ESM-IF1[12], just like ProteinMPNN, designs sequences that fit certain backbone traces with a sequence recovery of 51% (52.4% for ProteinMPNN). This model generalises to protein complexes despite not being trained using this information, although the performance for recovering interface residues is not known.

Here, we explore the possibility of designing peptide binders towards 2843 heterodimeric protein interfaces and 1172 homodimeric interfaces using ESM-IF1 generated sequences evaluated with AF. We utilise available structures to generate seed structures with Foldseek[17]. Together, our approach results in an automatic pipeline for designing peptide binders with the possibility to scale to a large number of different target proteins.

## Results

We begin with analysing the optimal way to predict the structure of protein–peptide complexes and how to select the binders which can be predicted with high accuracy. We find that AlphaFold2 can distinguish true binding residues and create a loss function that can select true binders from experimental data. We consider binders to be successful when predicted and native peptides have an interface RMSD ≤ 2 Å. This same loss function (Eq. 1) is used throughout the study to evaluate binders. We outline a procedure for binder design and evaluate it on all heteromeric interfaces in the PDB in a zero-shot approach. We continue investigating the criteria for successful binder design, representing highly confident predicted binders, convergence, and off-target effects, and compare the performance with other protein design methods. Notably, we apply methods developed only for single-chain applications enabling large-scale evaluations of generalisation abilities across known protein complexes.

## Determining and validating a loss function for protein-peptide design

*Optimal protein–peptide structure prediction.* AlphaFold2 (AF) can predict the structure of protein–peptide complexes using a multiple sequence alignment (MSA) to represent the target structure and the single sequence for the peptide representation[10]. The performance of this approach was reported using an increased number of recycles (nine) and as the best of ten different models. If one aims to search for a sequence using AF as a target function, having many recycles is ineffective, and a top-10 approach is unrealistic.

We evaluated the performance of AF on a set of 96 non-redundant peptides[10] using 1–10 recycles on top-1 predictions. Fig. 1a shows the cumulative fraction of peptides below a certain threshold in the average interface RMSD for different numbers of recycles. The interface is defined to contain all beta carbons (Cβs) within 8 Å between the peptide and its receptor protein. The interface RMSD was calculated after aligning the alpha carbons (Cαs) between the predicted and native structures. The same definition is used throughout the analyses here. We find that at above eight recycles, no improvement is observed, and 12.5% of the models ($n = 12$ out of 96) are within 2 Å RMSD. At one recycle, only six models are correct at 2 Å.

Since not all peptide sequences can be predicted accurately at their true locations, we analyse if we can distinguish when predictions are accurate. We analyse the predicted lDDT score (plDDT), a measure of how accurately AF predicts each protein residue and the distance to the target interface residues. The interface distance is calculated by taking the shortest distance from each atom in the target interface residues toward any atom in the peptide and then averaging.

Figure 1b (data in supplementary data) shows a ROC-curve where positives ($n = 12$; negatives, $n = 84$) have an RMSD below 2 Å towards the true peptide structure interface. The plDDT of the peptide results in the highest AUC (0.94), and combining the plDDT with the receptor interface dist results in a slightly higher true positive rate (TPR) at a low false positive rate (FPR) with no reduction in AUC, although many positives are missed (84 out of 96). Together, this suggests that although AF is largely inaccurate at predicting protein-peptide interactions, when it does so accurately the peptide will be situated close to the target interface and have a high plDDT score.

*AlphaFold2 can distinguish true binders from mutated ones.* AF may have learned certain binding pockets or potential interface regions, which would result in any sequence that could be predicted to interact with these regions. To analyse if it is possible to distinguish between potential peptide sequences to bind to a certain target interface, we mutate the 12 peptides that AF can predict at 2 Å RMSD (see above).

We randomly alter amino acids in contact with the receptor in the native peptide sequences and predict the structure with AF (using 8 recycles). For each number of contact residues, we introduce 10 additional sequences, resulting in 10·L combinations for each peptide (1160 sequences in total for the 12 peptides with an average length of 12 residues).

The peptides with low mutation fraction have high plDDT (Fig. 1c) with a strong concentration of samples at above 80 plDDT and below 20% mutations. The receptor IF distance tends to increase with the mutation fraction (Supplementary Figs. 1 and 2). The peptides with low mutation fraction and low average

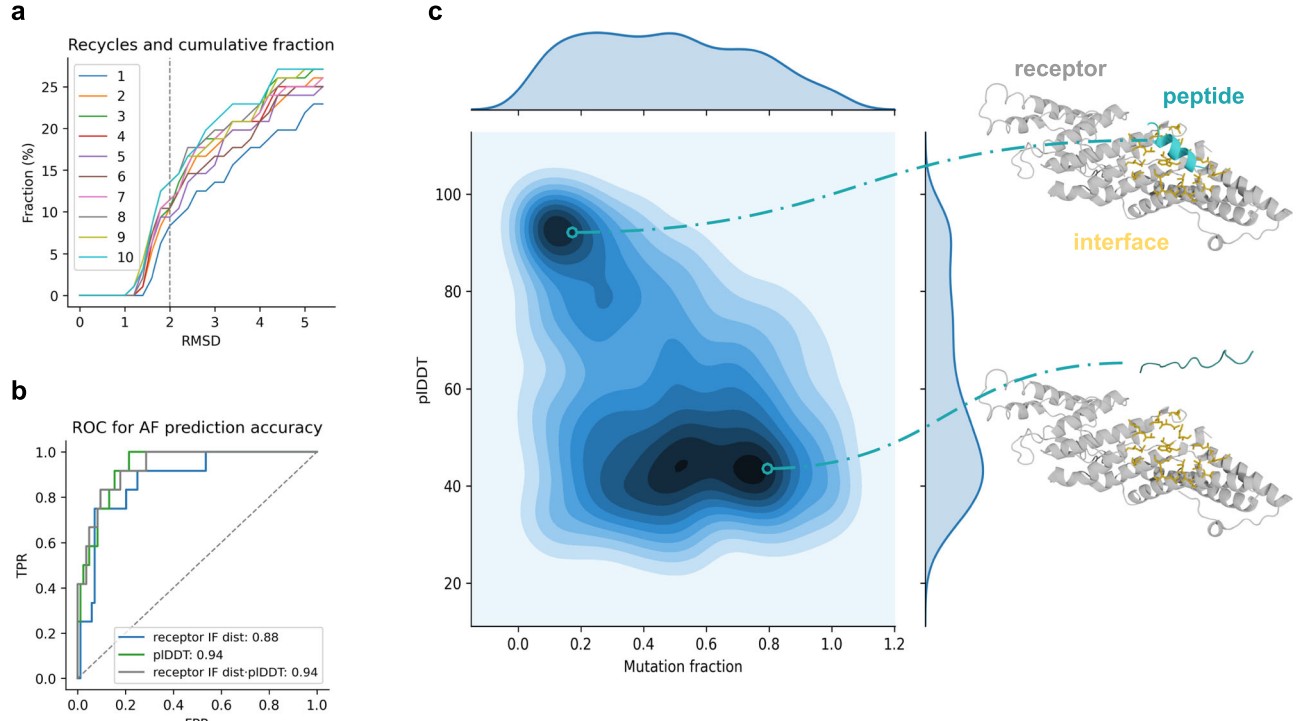

**Fig. 1 Overview of the AF prediction accuracy. a** Analysis of the effect of the number of recycles on the outcome, measured by the average peptide interface (IF) RMSD. The IF is defined as Cβs within 8 Å between the peptide and its receptor protein. The IF RMSD was calculated after aligning the Cαs between the predicted and native receptor structures. The cumulative fraction vs the RMSD threshold (cut at 5 Å) is shown. At 8 recycles and above, no improvement at 2 Å (grey dotted line) is found, although the difference between 1 and 8 recycles is 12.5% vs 6.25%. **b** ROC curve where positives ($n = 12$; negatives, $n = 84$) are predictions with an RMSD $\leq 2$ Å using 8 recycles. The plDDT of the peptide (plDDT) results in the highest AUC (0.94), and combining the plDDT with the distance from the receptor target residues to the peptide results in a slightly higher TPR at low FPR with no reduction in AUC. **c** plDDT vs the fraction of mutated IF residues compared to the total peptide length using the peptides that could be predicted at 2 Å RMSD ($n = 12$). In total, there are 1160 samples, 10 for each interface residue in the peptides. An illustrative example for PDB ID 3c3o (grey) is shown to provide intuition for this principle, where the peptide with a low mutation fraction (cyan, top) is highly ordered and close to the interface (orange). In contrast, the peptide with the high mutation fraction is disordered and further away (bottom). For data, see supplementary data 1.

distance to the receptor interface have high plDDT. It is also possible that the mutated peptides can still bind to the target interfaces, at least to some degree, which may explain why some mutated peptides are predicted close to the target interface.

However, AF seems to be less sure of the placement of these sequences, indicated by the lower plDDT (Supplementary Fig. 1). By selecting peptide sequences with a low average distance to specified target residues and high plDDT, it may be possible to retrieve true binders. Based on these criteria, we create a loss function (Eq. 1) to evaluate binders in subsequent analyses.

$$Loss = binderplDDT^{-1} \cdot \left( \frac{1}{m}\sum_{i=1}^{m}d_i + \frac{1}{n}\sum_{j=1}^{n}d_j \cdot \frac{1}{2} \cdot \Delta COM \right)$$

(1)

The loss is calculated after structural alignment on the target receptor protein. The *binder plDDT* is the average plDDT from AF over the binder, $d_i$ is the shortest distance between all atoms in the receptor target atoms and any atom in the binder ($m$ pairs in total), $d_j$ is the shortest distance between all atoms in the binder and any atom in the receptor target atoms ($n$ pairs in total), and $\Delta COM$ is the Cα centre of the mass-weighted distance between the native and predicted binders. The $\Delta COM$ is taken towards the seed structure and added here to ensure the designs target the desired area and not its mirror image. Obtaining the same loss on the opposite side of a protein is possible without it.

*Selecting true binders from experimental data.* Miniproteins are small protein scaffolds with sizes of up to 70 residues. Such

proteins have been designed to bind target interfaces with limited success[3]. To see if the modification of the AF protocol described here can distinguish these binders, we analyse sequences tested against four different receptor proteins with solved receptor-binder structures (FGFR2: https://www.rcsb.org/structure/7N1J, TrkA: https://www.rcsb.org/3d-view/7N3T/1, IL7Ra: https://www.rcsb.org/structure/7OPB and VirB8:https://www.rcsb.org/structure/7SH3)[3].

These binders were evaluated by counting the number of times they were detected to interact on yeast cell surfaces by Fluorescence-activated Cell Sorting (FACS), so-called Next-Generation Sequencing (NGS counts). We sampled 1000 sequences below 1000 NGS counts (to reduce the computational cost, Methods) and all above and predicted the receptor-miniprotein structures with AF using the same protocol described above.

Fig. 2 displays the relationship between the loss function and the normalised NGS counts (data in supplementary data). At NGS counts of zero, there is a higher tendency to obtain a high loss than compared to at 0.04, where the loss is close to 0. As a result, 20% of binders can be selected at an FPR of 10% using a success cutoff of 0.01 in normalised NGS counts (Supplementary Fig. 3). We note that this is not a very good ratio, although it does increase the likelihood of successful binder selection. N410mpg

Compared to the mutated peptide binders, the average plDDT is higher (84 vs 58), suggesting that AF is unsure about the residue locations of unbound peptides but not of unbound miniproteins. This corresponds well to the higher flexibility of

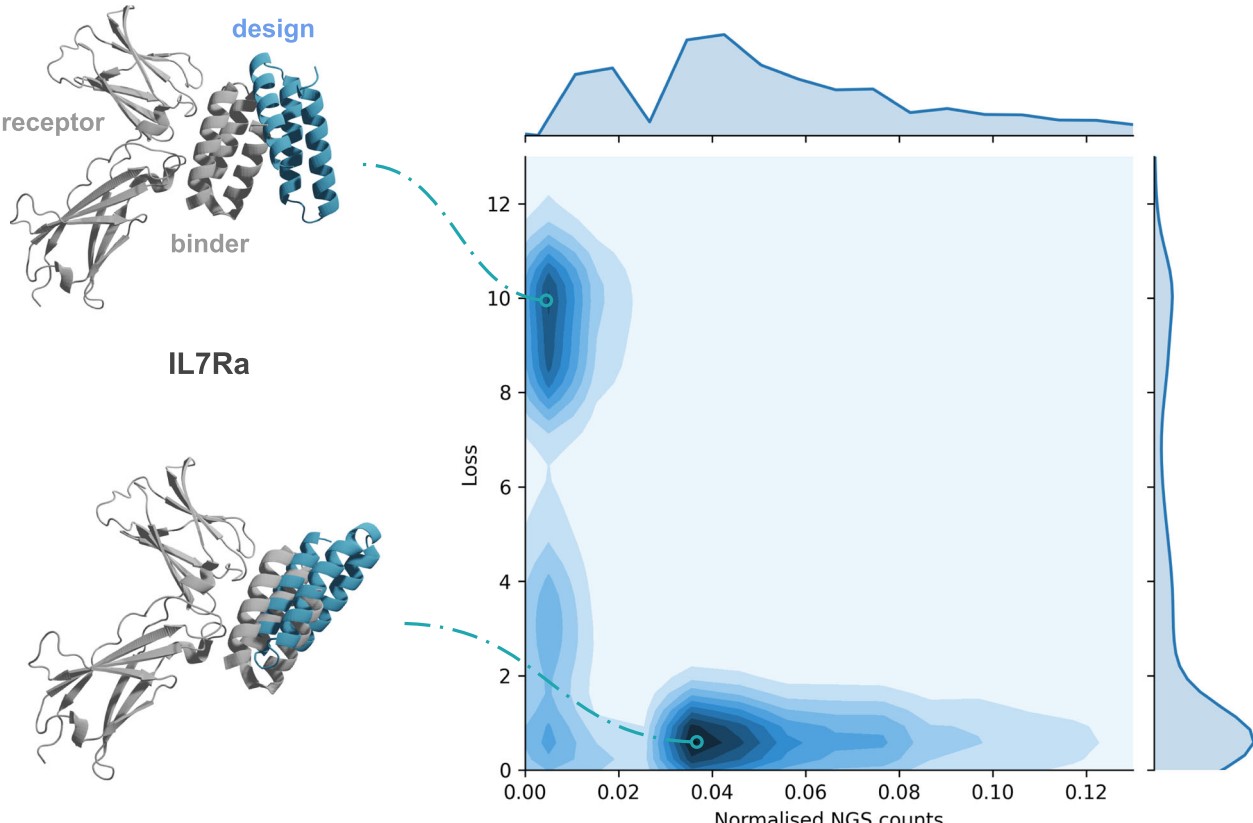

**Fig. 2 Loss and normalised NGS counts for the four tested systems of designed miniprotein binders with resolved structures (*n* = 5578 of which 2782 have NGS count 0).** An illustrative example of high and low loss for IL7Ra is shown with the resolved structures (native grey, design blue). The NGS counts are normalised by dividing with the highest count. The axis limits have been cut at 13 and 0.13 for the loss and normalised counts, respectively. For the full range, see Supplementary Fig. 4; for an ROC curve for selecting binders using the loss, see Supplementary Fig. 3. For data, see supplementary data 2.

peptides which likely only achieve their native configuration upon binding, highlighting that the plDDT alone is not enough to evaluate binders and the necessity for the loss function (Eq. 1).

*Experimental validation of the loss function.* The loss function developed here (Eq. 1) has been independently evaluated[18–20] for the target with PDB ID 1ssc [https://www.rcsb.org/structure/1ssc]. The study showed that the loss function is highly accurate for selecting true binders and compared Markov chain Monte Carlo (MCMC) search with Monte Carlo tree search (MCTS) for designing sequences. This provides support for the utility of the loss function and underlines that the problem of binder design is to find sequences that generate low losses, as suggested in Fig. 1c. Using MCMC search, two out of three designs have µM affinity and the best design out of three reported a $K_D$ of 2.5 µM (Table 1). With MCTS the best design had a $K_D$ of 0.08 µM and all selections had µM affinity.

**Binder design**. In total, there are 200,000 structures in the PDB[21]. We create a joint framework (Fig. 3) by searching all known structural information with Foldseek to generate seeds for the protein design method ESM-IF1 adapted to design receptor-binder sequences evaluated with AF. For data, see supplementary data 3. We evaluate our pipeline on 1463 nonredundant protein complexes corresponding to all (2926) unique heteromeric protein interfaces in the PDB. One seed per target is evaluated here, although many more seeds can be used. Importantly, ESM-IF1 and AF have not seen any of these complexes before, as they are trained only on single-chain proteins.

**Table 1 Affinity ($K_D$) for three different sequence selections designed towards PDB ID 1ssc [https://www.rcsb.org/structure/1ssc] using the loss function (Eq. 1) from another study[18].**

| Sequence selection | MCMC affinity ($K_D$) | MCTS affinity ($K_D$) |
|---|---|---|
| 1 | 2.5 µM | 0.08 µM |
| 2 | 8.7 µM | 0.12 µM |
| 3 | NA | 1.2 µM |

The sequences were designed with MCMC and MCTS, respectively.

We select continuous crops of 10–50 residues from each interaction partner, analogous to search results from Foldseek across the PDB (Methods). These crops are concatenated with the target proteins, adding a mask of 10 residues in between, and the backbones are used to design sequences with ESM-IF1. Only one sequence per target interface is generated to evaluate the capacity of the 'zero-shot' design. Fig. 4a shows the results evaluated with AF.

Our custom loss function, acting as a *bind score* (Eq. 1), can select binders with low interface RMSD and these have high contact recovery (>80%), suggesting highly successful designs. If AF was not a good evaluator, all sequences would end up at the interface (or none), and any residues could be put together, resulting in low contact recovery. That AF can distinguish between true binding sequences (proven experimentally[13–15]) is a remarkable generalisation capability resulting from the ability to predict protein complexes[4,6].

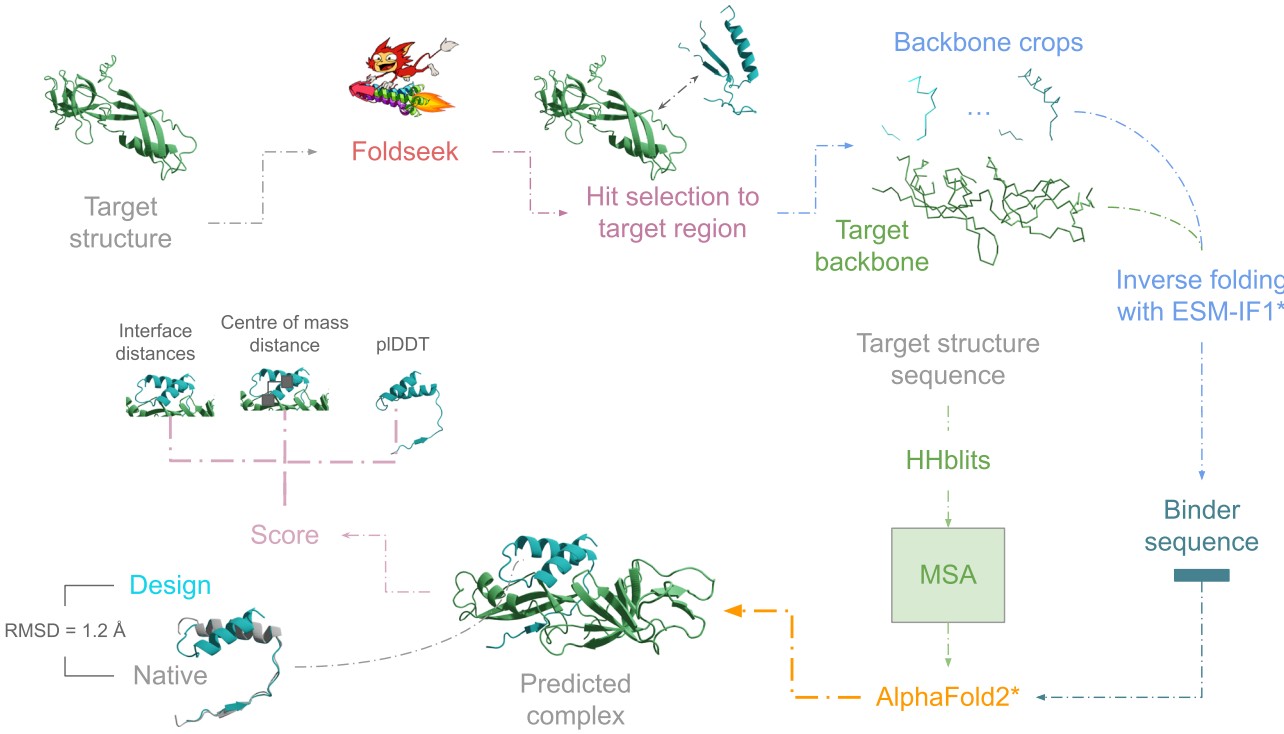

**Fig. 3 Outline of the design procedure.** A target structure (predicted/experimental) is taken as input and searched with Foldseek to produce hits. The hit with the highest contact density to the target interface is then selected, and inverse folding with ESM-IF1 generates a binder sequence. The target structure sequence is searched against Uniclust30 with HHblits, and the resulting MSA is used together with the designed sequence to predict a protein complex structure. The design is evaluated using the custom loss based on the interface distances, the centre of mass and the plDDT of the design. An example for 2RF4_A-2RF4_B is shown, with the final design towards 2RF4_A in structural superposition with the native seed. The RMSD in the interface positions is 1.2 Å between the native and designed binder. For data, see supplementary data 3.

The AUC ROC for selecting designs at 2 Å RMSD is 0.96 using the loss (Fig. 4b). At an FPR of 10%, 87% of successful designs (RMSD ≤ 2 Å) can be selected. The success rate is the highest (1.9%) for zero-shot designs using a length of 10 residues and decreases somewhat with length (Fig. 4c). This is likely due to the relative strength of the binding signal in shorter sequences (Fig. 4d).

Shorter sequences will depend more on inter-chain interactions to obtain their final folds, as many short peptides likely only adopt their structures upon interaction with their target receptors. Shorter designs have lower interface sequence recovery than longer ones (Fig. 4e). This indicates the importance of the evaluator function and the ability of AF to predict the designs with sufficient accuracy, something we analyse further below. In total, 205 successful designs were created towards 137 unique target interfaces.

**Binder design convergence.** The zero-shot analysis (Fig. 4) determined that a length of 10 residues is most likely to produce successful designs. Usually, more than one sequence is evaluated in protein design to improve performance. The sequence design's temperature (or noise) regulates the sampling, with low temperatures resulting in more deterministic behaviour. For the zero-shot analysis, a sampling temperature of $10^{-6}$ was used, resulting in almost deterministic sampling. We find that temperatures below 1 result in repeated designs using a sample size of 10 sequences, while at 1, there is an almost complete divergence for 100 sequences (Methods).

To analyse how many examples need to be generated to reach successful designs (binder interface RMSD ≤ 2 Å), we generate 100 diverse samples for each seed using a length of 10 residues

and sampling temperature 1. The average success rate plateaus at around 6.4% for 80 samples (Fig. 5a and Supplementary Data). The overall success rate using all designs is 6.5 %, resulting in successful designs towards 185 out of 2843 interfaces, a 3.4-fold improvement compared to the zero-shot approach (1.9%).

Using a loss cutoff of 1 with plDDT>80, 131 successful designs can be selected with a TPR of 95%. The relationship between the loss, RMSD, and fraction of recovered contacts is similar to that observed in Fig. 4a (see Supplementary Fig. 5). The Spearman correlation between the binder interface RMSD and the loss is 0.87. Sampling more sequences results in an improved success rate, and the accurate designs can be distinguished from the loss, just like for the zero-shot evaluation.

To see if AF can distinguish the interface contacts for these designs, we compare the sequence recovery in the interface residues of successful and unsuccessful designs for the same ids ($n = 185$). We find that the average plDDT increases with the binder interface sequence recovery (Fig. 5b) and that at sequence recoveries above 80%, the average plDDT is 84 and the average binder interface RMSD 0.43. This is consistent with the findings regarding the peptides with known structures in Fig. 1c, indicating that obtaining high plDDT values equals designs with low interface RMSD and high sequence recovery.

**Designed binder specificity.** When designing binders, binding only to a target interface and not to other proteins is desirable. To evaluate the specificity, we predict the interaction of the best-designed sequences from the convergence analysis that can be predicted at 2 Å RMSD ($n = 185$) with a set of 100 randomly selected receptor proteins from the remainder of the

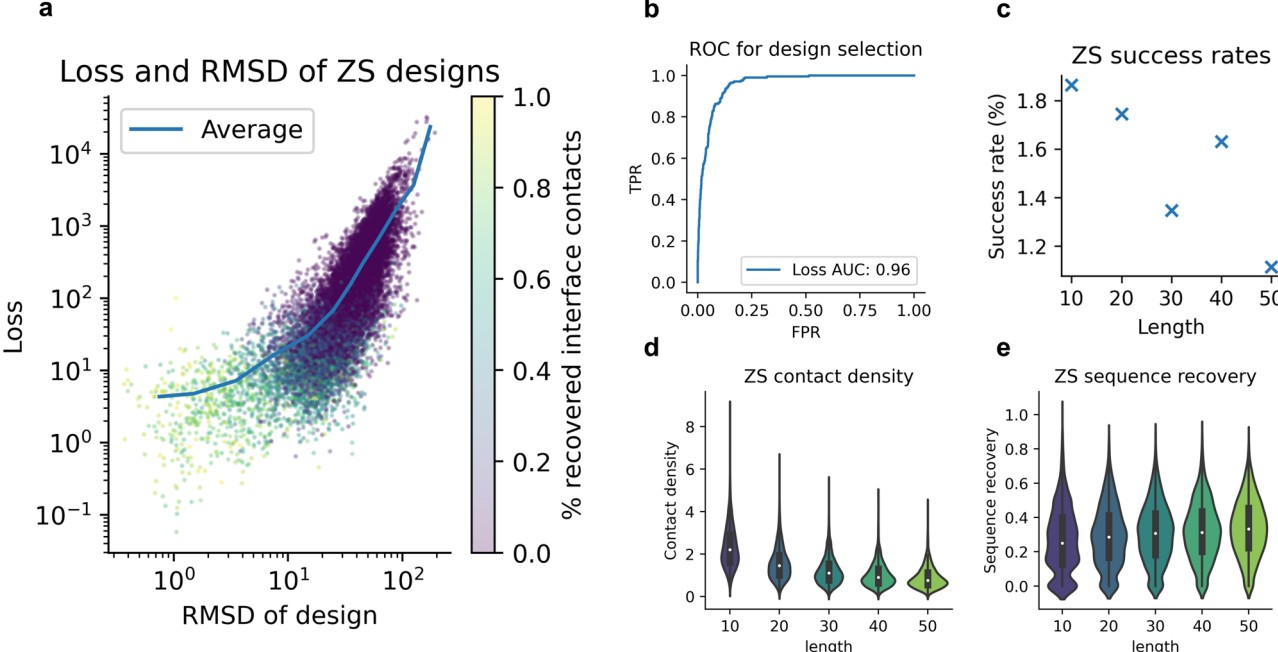

**Fig. 4 Analysis of zero-shot (ZS) designs using only one generated sequence per target interface-binder pair. a** Loss vs the RMSD in the interface between the designs predicted with AlphaFold2 and native binders ($n = 13,216$, Spearman $R = 0.81$). The points are coloured by the fraction of recovered interface contacts. When the loss is low, the RMSD of the design is low, and the fraction of recovered contacts is high (>80%). **b** TPR vs FPR for selecting designed binders with less than 2 Å difference in the interface compared to the native structures. Using the loss function, the ROC AUC = 0.96. At an FPR of 10%, 87% of successful designs can be selected (loss threshold = 0.11). **c** Success rate (binder interface RMSD ≤ 2) vs the length of each design ($n = 13,156$, n10 = 2895, n20 = 2805, n30 = 2672, n40 = 2451, n50 = 2333). The highest success rate is obtained for the shortest (10 residues, 1.9%) designs and the lowest for the longest (50 residues, 1.1%). In total, 205 successful designs are created for 136 unique target interfaces (see Supplementary Table 1 for the number of successful runs out of the number of possible runs). **d** Contact density vs length of the native binder crops. The shorter sequences have a higher number of contacts per position. This likely makes the interactions easier to predict, explaining the higher success rate. The black boxes encompass data quartiles and the white dots mark the medians. **e** Interface sequence recovery vs length. The shortest sequences have the lowest median sequence recovery, although the differences are only a few percent. The black boxes encompass data quartiles and the white dots mark the medians.

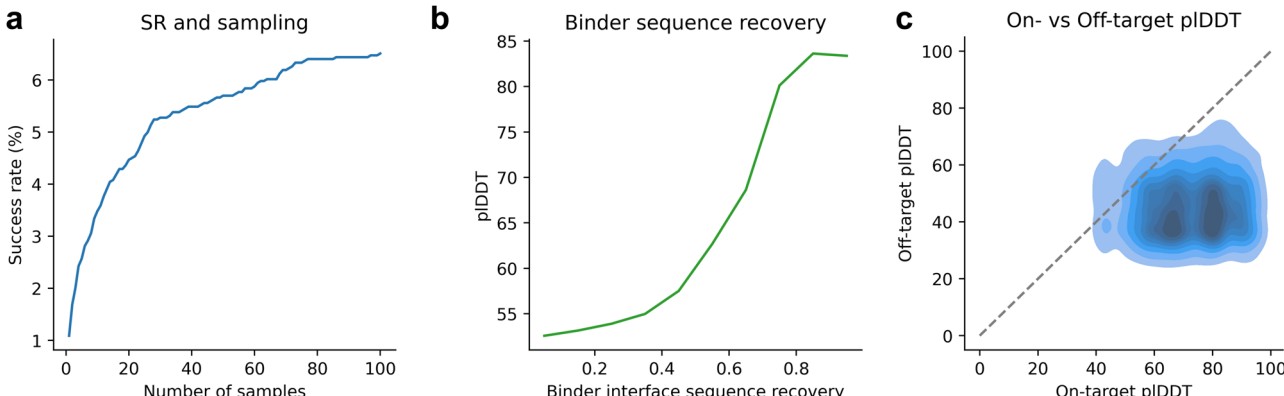

**Fig. 5 Overview of success rate and sequence recovery. a** Success rate (having one design ≤2 Å interface RMSD) vs the number of designs for a total of 2843 interfaces and 1–100 designs. The total success rate is 6.5 %, resulting in successful designs towards 185 unique interfaces (see Supplementary Table 1 for the number of successful runs out of the number of possible runs). **b** Sequence recovery (step size = 0.1) in the binder interface vs the average plDDT for the successful designs ($n = 18,500$). At low sequence recoveries, the plDDT is likely to be low (around 50). The average sequence recovery increases with the plDDT and at sequence recoveries above 80%, the average plDDT is 84. (see Supplementary Table 1 for the number of successful runs out of the number of possible runs). **c** Specificity analysis of the 185 successful binder designs towards 100 randomly selected receptor proteins ($n = 18,500$). The plDDT is consistently lower when predicting the binder in complex towards a random set of 100 receptor proteins than towards the intended receptor (On-target median plDDT = 72, Off-target median plDDT = 45). For data, see supplementary data 4.

nonredundant heterodimeric protein complexes. Even though the randomly selected proteins are dissimilar to the target proteins, the analysis helps to assess the specificity towards a certain class of interfaces.

Figure 5c shows the plDDT distribution for the true receptors (on-target) vs the random selection (off-target). The plDDT of the binders predicted in complex with the off-target receptors is consistently lower with values between 30 and 60, while the on-

target plDDT values are between 60–90. The median plDDT value is 45 for the off-target and 72 (60% higher) for the on-target binders. As observed in Fig. 1c, AF does not predict high plDDT values for interactions that are not intended. The binding to the intended target receptors can be selected with an ROC AUC of 0.96 using higher plDDT as the discriminator (TPR = 100% at FPR = 10%, see Supplementary Fig. 6).

**Analysis of failed designs**. We have shown that AF can evaluate binder sequences that can be predicted with high accuracy in previous sections. Therefore, the task of binder design is to generate a sequence that AF can predict. Successful designs can be generated towards 6.5% of the heterodimeric interfaces. To analyse what causes the failures (93.5%), we investigate if AF is better at understanding regions with defined secondary structures. Fig. 6a shows the fraction of secondary structure in the target receptor interface positions and the binder interface $RMSD^{-1}$. For data see supplementary data. There is no relationship between the interface and the RMSD, suggesting that there is no preference for designing a particular interface.

Other important aspects are the contact density in the interactions and the available evolutionary information in the target protein MSAs measured by the number of effective sequences (Neff, Methods). At high contact densities, the success rate is high (Spearman $R = 0.93$). This suggests that obtaining suitable seeds for the design process is the most important and is more likely for shorter sequences (Fig. 4d). At a contact density of 8.8, the success rate is 67% compared to <1% at a contact density of 1. We find no relationship between Neff and the RMSD, indicating that successful designs can be generated at high and low Neff values.

**Comparison with other protein design methods**. We compare the design success rate with the recent method ProteinMPNN[13], proven to substantially outperform one of the foundational methods for protein design, Rosetta[22]. So far, ESM-IF1 has been evaluated on the harder problem of heteromeric interfaces (as homomeric sequences may be inferred from the receptor target proteins).

Most of the examples in the test set of ProteinMPNN are homomeric (99.3%, $n = 677$, Methods). These are used to compare 100 sequences sampled for each method using a seed length of 10 (Fig. 6f). In total, 100 designs, each towards 1172 interfaces were evaluated. The success rate (interface RMSD ≤ 2 Å) is 3.6% for ESM-IF1 and 1.5% for ProteinMPNN, resulting in 42 and 18 accurate designs, respectively. ESM-IF1 thereby outperforms ProteinMPNN by a factor of 2.3 in success rate.

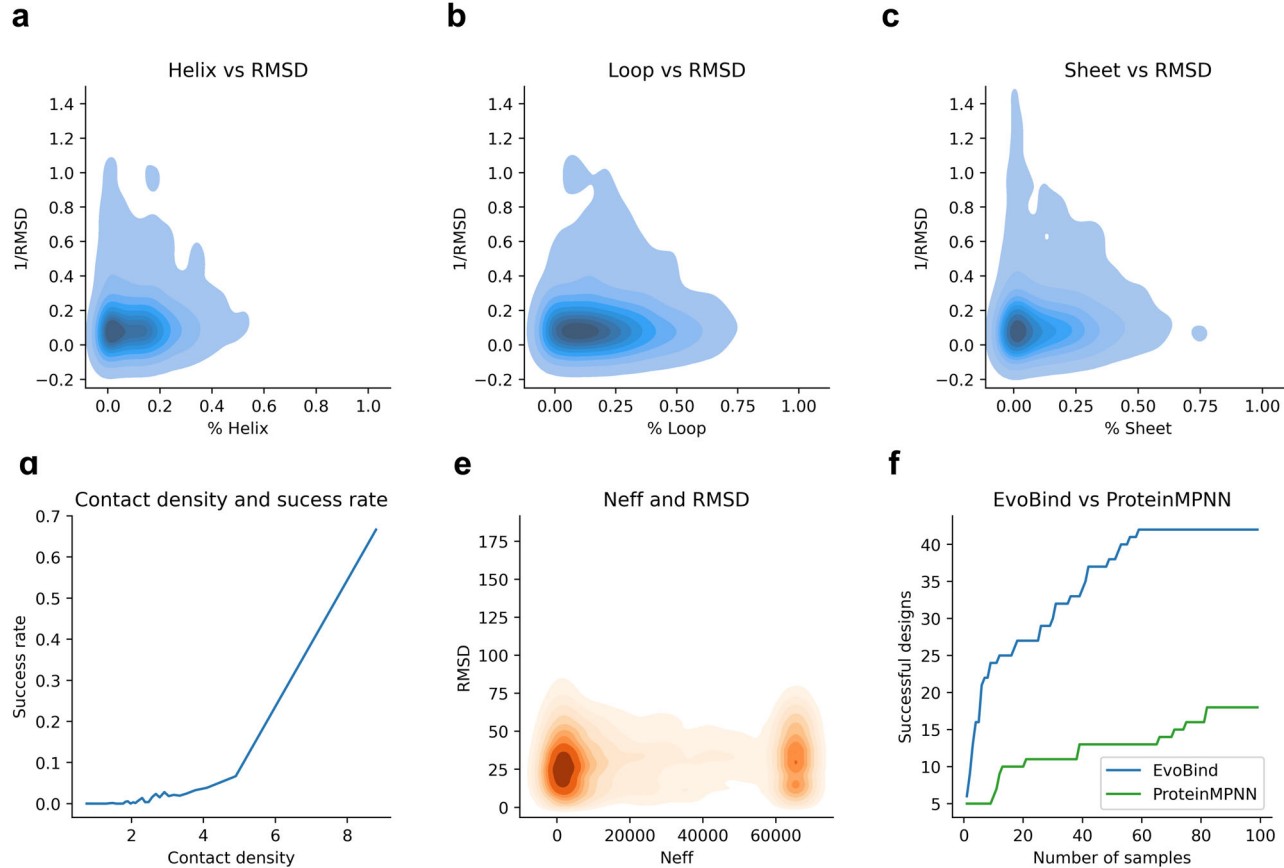

**Fig. 6 Overview of success rate for different structural groups. a–c** Receptor interface type fraction vs the interface RMSD-1 compared with the native structures. High values mean low RMSD here since the RMSD is inverted. There is no relationship between the interface and the RMSD, suggesting that there is no preference for designing a particular interface. **d** Contact density (number of contacts per residue) and success rate using all designs ($n = 280,553$, SpearmanR = 0.93). The data was divided into 30 even partitions ($n = 9345$ per partition) after sorting on contact density, and the average contact density vs the success rate was taken for each partition. At the highest contact density (8.8), the success rate is 67%. See Supplementary Table 1 for the number of successful runs out of the number of possible runs. **e** Number of effective sequences (Neff) of the MSAs and the RMSD of the designs. There is no relationship between Neff and RMSD. **f** Comparison of the number of successful designs (binder interface RMSD ≤ 2) from ProteinMPNN compared to ESM-IF1 using 1–100 generated sequences per target interface ($n = 1172$ interfaces). The success rate is 3.6% for ESM-IF1 and 1.5% for ProteinMPNN resulting in 42 and 18 accurate designs, respectively. For data, see supplementary data 5.

## Conclusions

We have shown that AlphaFold2 can distinguish true binders from mutated ones and created a loss function that can select true miniprotein binders from experimental data consistent with recent findings[10,13–15]. Evaluating against all nonredundant heteromeric protein interfaces in the PDB, successful binders (≤2Å interface RMSD) can be selected with an AUC ROC of 0.96 in a zero-shot approach (one design per target).

With 100 generated sequences per target, ESM-IF1 can design successful binders (≤2 Å interface RMSD) towards 185 (6.5%) of all known heteromeric protein interfaces. We find that the designed sequences are specific toward the classes of receptor interfaces, represented by median plDDT values of 72 vs 45 towards unintended targets. When generating designs, we find no preference for interface types or the MSA depth (Neff). The contact density is the determining feature for the design success rate, and if one can obtain scaffolds with high density, the design success rate can increase from below 1% to 67%.

ESM-IF1 reports a success rate of 3.6% vs 1.5% (42 and 18 successful designs, respectively) for ProteinMPNN on 1172 homomeric interfaces (ProteinMPNN's test set). The expected performance across heterodimeric binder domains is thereby almost double of that on homomeric ones.

The rapid increase of AI technology has led to a revolution in the field of structural biology and protein design. Still, many challenging tasks remain as accurate binders are not produced for most interfaces here (93.5% and 96.4% failure rates for heterodimeric and homodimeric interfaces, respectively). This issue may be addressed by obtaining better scaffolds, as we have shown that if scaffolds with high contact densities can be generated, designs are more likely to be successful.

We expect that if a scaffold with a high contact density and low loss is obtained, accurate designs can be generated with ESM-IF1, and these can be selected with high confidence. A factor that may impede the design is the possibility of predicting unintended conformations, which depends on the ability of AlphaFold2. As more structures are being solved at an accelerated pace, we do expect that better scaffolds will be available in the future, including for multiple conformations.

## Methods

**Peptide structural data**. The peptide dataset was taken from a recent study[10] where 96 non-redundant protein-peptide structures were extracted from the PDB and manually analysed to ensure structural divergence (each involves a distinct fold). The details for creating this dataset can be found in the original publication[10]. The binders designed here have a maximum length of 50 residues.

**Miniprotein binders**. Four different designed miniprotein binders that had resolved structures towards single-chain proteins were selected from a recent study (FGFR2: https://www.rcsb.org/structure/7N1J, TrkA: https://www.rcsb.org/3d-view/7N3T/1, IL7Ra: https://www.rcsb.org/structure/7OPB and VirB8: https://www.rcsb.org/structure/7SH3)[3]. These binders were evaluated by counting the number of times they were detected to interact on yeast cell surfaces by Next-Generation Sequencing (NGS counts) in subsequent pools obtained from Fluorescence-activated Cell Sorting (FACS). We used the final pools for each of the four receptors as the designs there will represent the strongest binders. In total, there were 172,581 sequences available in this study and to limit the computational requirements, we sampled up to 1000 sequences below 1000 NGS counts and all above and predicted the receptor-miniprotein structures with AF as described above. In total, 5578 receptor-miniprotein structures were

evaluated, 2013 for FGFR2, 1099 for TrkA, 1249 for IL7Ra, and 1217 for VirB8 (see Supplementary Fig. 7 for the NGS count distributions). For more details, see Supplementary Notes.

**Structure prediction with AlphaFold**. A modification of AlphaFold (v2.0)[23] (AF) based on the FoldDock protocol[4] and a recent study for protein-peptide structure prediction[10] was run, where the receptor is represented as an MSA and the peptide as a single sequence. The MSA was constructed from a single search with HHblits[24] version 3.1.0 against uniclust30_2018_08[25] using the options:

hhblits -E 0 001 -all -oa3m -n 2

The MSA of the receptor was input together with the single sequence representation of the peptide (binder) to the AF folding pipeline, using model_1, one ensemble and between 1 and 10 recycles (8 were found to be optimal, see Fig. 1a). In model_1, no predicted TM-score or predicted aligned error is available, only the predicted lDDT (plDDT) for each residue. We used model_1 since this has been found to be optimal for predicting heterodimeric complexes, which is a highly related task[4].

The structural prediction was performed on one NVIDIA A100 Tensor Core GPUs with 40 Gb of RAM. On average, compiling the folding pipeline took 144 s and each iteration took 46 s, resulting in an average total time of 144 + 46·number of designed sequences. The designed sequences were evaluated using AF by predicting the interaction with the receptor structure and calculating the loss (Eq. 1).

For the zero-shot designs, 13,156/13,893 designs (95%) could be predicted successfully and for the convergence analysis, 282,853/292,600 (97%). The failed ones were due to inconsistencies in the input pipeline generated by MSA sampling leading to shape mismatches. We deem the failed examples too few to impact the overall results.

For the ProteinMPNN set (see below), predictions were produced towards 1172 interfaces with 115,855/117,200 designs for ESM-IF1 and 116,334/117,200 for ProteinMPNN. The missing examples were due to time limitations running 8-h jobs with resources as specified above. See Supplementary Table 1 for the number of successful runs out of the number of possible runs.

**Unique pairs of Pfam domains (heteromeric set)**. The main dataset for evaluation was taken from a recent study[26]. This set consists of 1661 protein complexes that have a resolution <5 Å, constitute a unique set of interacting Pfam domains, and share <30% sequence identity. This dataset has been used previously[4] to evaluate AF and AFM for protein-structure prediction of complexes, where some structures were removed for various reasons (some lack beta carbons, backbone atoms or are large assemblies), resulting in a total of 1463 structures.

**Seed generation**. To generate backbone seeds for ESM-IF1, we select continuous regions of 10–50 residues with steps of ten residues where the contact density of the continuous region is the highest.

For some structures, crops of 50 cannot be selected (they are too short). In these cases, we use as many residues as possible. Since Foldseek will return the nonredundant set from the PDB (or complexes with very high similarity), we skipped this step in the design evaluation here. The principle applies to any interface region:

1. Search PDB using the target structure with Foldseek (default settings).
2. Parse all results and select those with contacts to the defined interface area.

3. Rank the hits by the number of contacts within a continuous segment of length L corresponding to the desired design length.

4. Select the highest-scoring hit and extract (crop) the backbone of the interface segment to be used as a seed in ESM-IF1.

**ESM-IF1**. In the ESM-IF1 preprint, complex prediction is suggested as follows:

"Although the training data only consists of single chains, we find that models generalise to multi-chain protein complexes. We represent complexes by concatenating the chains together with 10 mask tokens between chains and place the target chain for sequence design at the beginning during concatenation"

Placing the target chain first (being the binder in this case) results in a high bias for putting a Methionine in the first sequence position. Out of 13893 designs in total with the zero-shot approach, 6700 have M in pos 1 (48%). ESM-IF1 has thereby learned to put M in pos 1, due to this being the most abundant starting position in proteins. The corresponding Methioinine frequency for the true sequences is 3%. When designing binders, the seed backbone for the binder should thereby not be concatenated first. Instead, we concatenate the target receptor backbone, followed by a masked region (np.inf) of 10 residues followed by the binder seed. This reduces the Methionine frequency to 0.1%.

We find that when analysing the temperature, a temperature of 0.1 generates 66% unique sequences, 0.5 99%, and 1 99.99% across ten different sequences. We use a temperature of $10^{-6}$ for zero-shot designs (deterministic) and 1 for the convergence analysis (100 different sequences per seed, 99.9% diversity). For some sequences, ESM-IF1 produces the character <eos>. The cases for which this occurred were disregarded.

In total, 13,156/13,893 designs were generated for the zero-shot evaluation and 282,853/292,600 for the convergence analysis (100 per unique interface) using the heteromeric set. For the ProteinMPNN test set of homologous proteins (see below), 65,400/67,700 designs were generated. The failed designs were either due to missing backbone coordinates or due to generating the character <eos>.

**Contact recovery**. To analyse how similar the designed binders are to the native ones, we compare the contacts (defined as beta carbons (Cβs) within 8 Å from each other) between the receptor and the native binders and of the receptor and the design. We group amino acids into five different categories (see below) depending on their physical characteristics to capture the fact that many different amino acids may interact with receptor interface residues equally well (e.g., different positive residues). From these groupings, we calculate the fraction of interactions in the native binder that is preserved in the design, which we call contact recovery.

What matters most are the residues in contact with the target interface, not sequence recovery, as the order of residues can differ by one position and still interact with the target interface. For each position in the native receptor interface, the interacting residues are extracted, and the unique groupings are annotated. This means that repeat contacts are not counted, e.g. if residue one interacts with A, F, and R, this translates to Hydrophobic and Positive. The two counts of hydrophobic residues (A and F) are not considered. This is because contacts may vary, and it may be sufficient to have one stronger hydrophobic interaction as compared to two weaker ones.

*Amino acid categories*. Hydrophobic: A, F, I, L, M, P, V, W, Y
Small: G
Polar: N, C, Q, S, T

Positive: R, H, K
Negative: D, E

**Interface sequence recovery**. To analyse the sequence recovery of the designs, we focus on the interface residues according to Fig. 1c. We extract the interface residues in the native sequences and calculate the identity in the same positions in the designs, assuming a linear sequence of residues from 1 to $N$ (the number of native interface residues). The interface sequence recovery is the number of identical residue-position combinations divided by $N$.

**Contact density**. To calculate the contact density, we start by extracting all beta carbons between a target receptor protein and a binder within 8 Å in a given seed. The total number of contacts is then divided by the length of the binder itself, resulting in a measure of the number of contacts per amino acid in the binder, which we name the contact density.

**Number of effective sequences**. To calculate the number of effective sequences (Neff), we cluster the MSAs from HHblits (above) at 62% sequence identity with MMseqs (version fcf52600801a73e95fd74068e1bb1afb437d719d)[27] with the options:

```
mmseqs easy-cluster --min-seq-id 0.62 -c 0.8
--cov-mode 1
```

The Neff values were calculated by taking the number of clusters at 62% sequence identity. In total, 2636/2645 alignments were clustered successfully.

**DSSP secondary structure annotation**. DSSP (version 2.0)[28] was run on all 1463 extracted pairs in the Pfam set. The secondary structure annotations were parsed into their respective classes (Helix, Sheet or Loop) and mapped to the interfaces of the target receptor-binder complexes. The states of all interface residues were counted to obtain the fraction of each secondary structure state in each interface.

**ProteinMPNN**. ProteinMPNN was trained on assemblies in the PDB downloaded on Aug 02 2021. These had resolutions below 3.5 Å and were determined by X-ray crystallography. The sequences were clustered at 30% identity, and 1539 clusters were saved for testing. The test data can be downloaded from: https://files.ipd.uw.edu/pub/training_sets/pdb_2021aug02_sample.tar.gz. When mapping these to unique PDB IDS, one finds 4086 PDB IDS in total. These contain 4019 chains mapping to 1117 unique cluster combinations, of which 1109 are homomeric and 8 heteromeric. Many of these complexes are not real interactions but only inferred from asymmetric units. Therefore, we fetched the first biological assemblies from all PDB IDs and extracted interacting pairs (one per PDB ID) based on having at least ten contacts (beta carbons <8 Å), resulting in a total of 677 pairwise interactions. For ESM-IF1, 654 of these produced designs (the rest contained missing regions in the backbone) leading to a total of 1308 possible target interfaces.

ProteinMPNN was run using its backbone weights (--ca_only) with a temperature of 1 (the default of 0.1 gave little sequence diversity) to generate an unbiased comparison with ESM-IF1, as there is also an option to use beta carbons and oxygens to design sequences with ProteinMPNN. 100 sequences were designed per interface, selecting the interface regions as target positions for the design according to the example provided in the ProteinMPNN GitHub repository: https://github.com/dauparas/ProteinMPNN/blob/main/examples/submit_example_4_non_fixed.sh

**Reporting summary**. Further information on research design is available in the Nature Portfolio Reporting Summary linked to this article.

## Data availability

All results and information required to reproduce this study are available from: https://gitlab.com/patrickbryant1/binder_design. All predictions are available from https://zenodo.org/record/8408449 (https://doi.org/10.5281/zenodo.8408449). Data for Figs. 1–5 is available as supplementary data.

## Code availability

All code required to reproduce this study are available from: https://gitlab.com/patrickbryant1/binder_design and a snapshot of the code used is uploaded as Supplementary Software. This is a pipeline for designing peptide binders- Binder design using a combination of [Foldseek](https://search), [ESM-IF1](https://www.biorxiv) and [AlphaFold](https://www). Foldseek is available under [GNU GPL-3.0](https://www.gnu.org/). ESM-IF1 is available under the [MIT license](https://opensource). AlphaFold2 is available under the [Apache License, Version 2.0](http://www.apache.org/). The AlphaFold2 parameters are made available under the terms of the [CC BY 4.0 license] and have not been modified. The binder design pipeline here is available under the same licences as a derivative of these methods.

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

## Acknowledgements

All protein structures were visualised using Blender (https://www.blender.org/). Financial support: Swedish Research Council for Natural Science, grant No. VR-2016-06301 and Swedish E-science Research Centre and from Knut and Alice Wallenberg Foundation. Computational resources: Swedish National Infrastructure for Computing, grants: SNIC 2021/5-297, SNIC 2021/6-197, Berzelius-2021-29 and Berzelius-2022-106. A.E. received all financial support and computational resources.

## Author contributions

P.B. designed and performed the studies and analyses. P.B. wrote the first draft of the manuscript and prepared all figures, which were later edited and improved by A.E. and P.B. A.E. obtained funding.

## Funding

## Competing interests

P.B. is the CEO of Urgenta Labs, a startup that develops targeted peptide binders. A.E. does not declare any competing interests.
