## [Peer Review File · Communications Chemistry]

This manuscript has been previously reviewed at another Nature Portfolio journal. This document only contains reviewer comments and rebuttal letters for versions considered at Communications Chemistry.

Reviewers' comments:

Reviewer #1 (Remarks to the Author):

The manuscript by Bryant & Elofsson introduces a protocol on top of AlphaFold for the design of peptide binders of a target. The subject is topical and timely, and generally, I believe this could be a nice contribution. The authors cover binder design issues quite extensively, including specificity, which is nice.

However, I have several remarks, some more about form than substance.

1) I did not find the manuscript so easy to read.

1.1) The manuscript is sometimes rather technical and several sentences/sections could be reformulated to get simpler. It is for instance the case of the second paragraph of the zero shot binder design section. I am not sure a standard reader could understand what is exactly done and why. Even I am unsure of the definition of a binder in this part. Is it just a fragment in contact with the target optimized using ESM-IF1? Is there any experimental validation of the binding? Please clarify. The sentence "In total, 205 successful designs were created towards 137 unique target interfaces" might be overstated.

1.2) At some places, additional information could be welcome. Among others: (i) all symbols of Equation 1 are not defined, (ii) I could not find in the manuscript the definition of the NGS counts, etc. I really believe that a slight rewriting of the manuscript could greatly improve the impact for the reader.

2) The interest of not having - in my understanding - any control on the region of the receptor targeted by the binders is not obvious - zero shot design section. Usually, one wishes to target one specific region, and have control on it. I suggest the authors could elaborate a bit on the pros and cons of such strategy. Also, could the fact of not controlling the interacting regions result in designs that could be easier to tackle?

3) To me, the section about binder specificity is not really convincing. Off-target issues often result from interactions with similar proteins, whereas here, the authors analyze the binding to unrelated proteins.

4) Some context about the perspective of having automated peptide binder design could be welcome. Having probes able to bind the target *in silico/in vitro* is really far away from something usable *in vivo*.

Reviewer #2 (Remarks to the Author):

I have written my comments in blue coloured text within the remarks threads of the reviewer comments pdf document attached.

Reviewers' comments:

Reviewer #2 (Remarks to the Author):

The authors report a computational pipeline for peptide binder design that combines three existing methods into a unified framework: a) determining initial structural interface similarity from a dataset of native heteromers (using Foldseek), followed by extraction of the corresponding backbone scaffolds as seed generators, b) sequence design from the chosen fixed backbone seeds (using ESM-IF1) and c) protein structure prediction (AlphaFold2 -AF2) from which a loss function is developed as a differentiator of binders from non-binders. The loss function combines output metrics from AF2 (plddt) with distance and centre of mass metrics and is validated against an experimental data set that counts the number of times binders interact on yeast cell surfaces by FACS and NGS counts. Computationally successful designs are selected as those that additionally exhibit an RMSD within 2Å of the structures from the native data set. The authors describe a zero-shot (ZS) design approach in which only one sequence is designed per target interface, followed by a study on convergence when using multiple sample design sequences reporting improvement over the ZS approach. They additionally try to improve predictions from the consensus of many designs, assess off-target effects, analyze failed designs in the context of several potential dependencies and finally compare their method with the recent ProteinMPNN design method.

In summary, the approach is an elegant one for peptide binder design, especially the use of Foldseek to extract backbone scaffolds from the initial data set as well as the benchmarking of the loss/scoring function against available experimental data. However, the study has several major shortcomings that I would like the authors to address before I could recommend this manuscript for publication. **The authors have performed many analyses, but some at quite a superficial level.** It would be more beneficial to the readership if the authors concentrated on the key aspects of their pipeline – **optimizing/varying their loss function**, exploring the rich content of multiple seeds from FoldSeek and re-designing their study to include a much larger sample size (depth) for a select subset of targets. This could better bring out the merits of their method, leaving other tangential analyses for the supporting information. Below I address these aspects inter alia in detail on a point-by-point basis:

We thank the reviewer for these great suggestions. Including another loss function would make this analysis very expensive computationally. Although this may perhaps be interesting to some readers, we believe the main interest is finding a loss function that is functional, which we believe we have demonstrated throughout the study. The same argument is made for increasing the depth. Although one could create 1000s of designs, we feel there is very little practical utility for this as a user will not want to do this in practice. Therefore, we limit the number of samples to 100. Using multiple seeds in FoldSeek is something that is implemented in the pipeline for new designs. We did however choose to limit the design process to only one seed per interface - again to limit the computational costs as we do not have unlimited resources.

The authors have in large part addressed my comments. However, a couple of the major points have not been covered satisfactorily.

The main outstanding issue in my opinion is that the manuscript would improve from the authors considering better what many 'users' want. Peptide discovery programs are significantly (although not exclusively) target driven. That is, users have in mind what targets they want to discover peptides/drugs for and then want a set of putative candidates from any given *in silico* program. So of-course a general method with wide applicability/coverage across multiple targets would be very useful for multiple users but it has to actually 'work' for any user-given target. Showing that it can maybe come up with one or two experimentally unvalidated *in silico* designs each across a subset of ~200 targets from a pool of ~3000 does probe the generality and so the authors are right to pursue this. But it doesn't demonstrate that the method gives experimentally testable pragmatic results for any given target. That is why depth analysis is important. Because a user will apply the method for their target(s) of choice, want to design 100s or 1000s of sequences *in silico* and then they or others may test those in the wet lab and perhaps only a handful of those will be experimentally positive.

A second issue is that while the authors have explained well their reasoning to most comments, these concepts have not been transmitted into the manuscript. In my opinion the manuscript and the readership would benefit if these concepts were discussed in summary in an expanded 'Discussion and conclusion' section.

I encourage the authors to address these issues before I could recommend the manuscript for publication.

Below I elaborate on a point by point basis:

Major points:

1. The authors could re-frame the first three sections of Results: 'Optimal protein-peptide structure prediction', "AlphaFold2 can identify binding residues in peptides" and "Selecting true binders from experimental data". These would read better as subsections in a section that focusses on determining and validating a loss function for protein-peptide binder design. The main point of the endeavor here, although not clear enough from the manuscript, is to come up with a loss function that is sufficiently accurate to discriminate a true structural binder, equivalent to <2Å RMSD to native binders but one which can then dispense for the need of RMSD comparison using native binders – thus enabling new designs to be evaluated. The subsections could then form a pathway to establishing this.

We thank the reviewer for this suggestion and have reformatted the manuscript to contain a section in the results called "Determining and validating a loss function for protein-peptide design" with subsections as suggested.

Addressed.

2. The subsection 'Optimal protein-peptide structure prediction' could be re-written more clearly. The authors should clarify that the standard AF2 structure prediction models do not alone give very accurate RMSD predictions for protein-peptide interfaces - only 12 out of 96 binders are predicted at their true location - even after optimizing the number of recycles. Nonetheless, they are still able to distinguish the binders that AF2 predicts to bind using the plddt score from AF2 (and coupled with their IF distance) – thus they have a good handle on

true positives albeit with many false negatives (which should also be commented on in the manuscript).

We will make the false-negatives notion more clear. We have added the following:

“The pLDDT of the peptide results in the highest AUC (0.94), and combining the pLDDT with the receptor interface dist results in a slightly higher true positive rate (TPR) at low false positive rate (FPR) with no reduction in AUC, although many positives are missed (84 out of 96). “

It would be good to emphasise this in the following sentence too, e.g. “Together, this suggests that although AF is largely inaccurate at predicting protein-peptide interactions, when it does so accurately the peptide will be situated close to the target interface and have a high pLDDT score.”

3. In the section “AlphaFold2 can identify binding residues in peptides” the authors show by mutating the 12 accurately predicted peptides that pLDDT decreases as mutation fraction increases and that at lower mutation fractions high pLDDT is more indicative of low IF distance. It is not clear which data suggests that AF2 can identify specific binding residues and the title and claims are somewhat misleading and should be changed. They should change the title of the section to something like: “AlphaFold2 can distinguish true binders from mutated ones” and also the claims in the conclusion.

We have changed this as suggested.

Addressed.

a) The authors could consider plotting the data from Supp Fig 1 by binning by mutation fraction, calculating the correlation between pLDDT and IF distance and plotting correlation vs mutation fraction bin. This would be insightful for the main manuscript as a new Fig 1 d.

Binning and then taking correlations introduces severe bias to the chosen bins resulting in finding correlations where there are none. We believe the main points of showing that 1. AF can predict some peptide binders with high accuracy and 2. these can be distinguished are portrayed clearly which is why we put this figure in the supplementary information.

Perhaps the term binning caused confusion. I will clarify. For each value of mutation fraction, determine the correlation between IF and pLDDT for the corresponding vertical line of points. Then plot correlation against mutation fraction. If what the authors state is true correlation should decrease with mutation fraction and this would in my opinion be easier to interpret rather than multiple columns of dots.

b) The authors use this process to develop a loss function combining pLDDT, IF distance and centre of mass. Does the complete loss function improve over the combination of pLDDT and IF distance in Fig 1b? They could add this to the plot.

The introduction of centre of mass is to avoid reflexions. Without it, it is possible to obtain the same loss on the opposite side of a protein and thereby not target the goal area. We have made this more clear in the text:

“The Δ COM is added here to ensure the designs target the desired area and not its mirror image. Without it, it is possible to obtain the same loss on the opposite side of a protein. “

Addressed.

4. Why do the authors choose the exact loss function they do and how does it compare with other potential choices? Although rationale is provided for the plddt and the IF distances, it is not for the COM – and indeed many combinations of terms are possible (and easily explorable once structures have been predicted). The authors could assess several additional loss functions and rank them. Why did the authors choose AF2 model_1 instead of the further fine-tuned model_1_ptm, which would provide the option of using the predicted aligned error (pAE) as a way of scoring binder candidates and that could potentially be integrated into more effective loss functions?

We added the COM to avoid reflexions as we observed in initial studies that these may introduce errors by designing to undesired areas, mentioned above. There are other options to consider. The reason for choosing model 1 is that we observed this one to be the best for protein complex prediction (<https://www.nature.com/articles/s41467-022-28865-w>) which is a highly related problem. We don't have the capacity to explore all suggested loss functions and instead leave this for future studies showing here only the performance of the one we designed.

Even though not explicitly computed in this study, the concept of using other loss functions, their potential advantages/disadvantages computational costs etc. should be discussed more thoroughly in the conclusions.

5. The validation of the loss function against the experimental miniprotein binder data set is an elegant idea. A few points should be considered here:

a) It would help to plot the 1000 NGS count threshold here on Figure 2. Although Figure 2 does show a strong transition from high to low loss on an increase in NGS counts, there seems still to be a significant false-positive (lower-left corner) distribution. Could this be explained by the reports that AlphaFold has been shown to predict inflated pLDDTs for short spurious proteins (see e.g. Monzon, Haft & Bateman 2022 doi.org/10.1093/bioadv/vbab043 for results on AntiFam). If so, do the authors have a strategy to potentially mitigate this issue?

This is a good point and we have analysed the FPR along with the inflated pLDDTs by taking the examples with normalised NGS counts < 0.01 and a loss < 1. These have a median plddt of 85.0, which is slightly higher than the median of all examples with normalised NGS counts < 0.01 (83.6). Note that all pLDDTs are quite high in this case, which is a characteristic of sequences designed with classical methods such as Rosetta - they are more stable and therefore easier to predict in single sequence mode. We have added to the section regarding Figure 2: “We note that this is not a very good ratio, although it does increase the likelihood of successful binder selection”.

The NGS count threshold was set to 0.01 after normalising with the highest count for each protein-binder pair. The 1000 NGS counts refer to the sampling (1000 sequences below 1000 NGS counts) used to reduce the computational requirements as there were 172581 sequences available from this study in total. Plotting the 0.01 selection threshold in Figure 2 we do not think is necessary as the densities at high/low loss/NGS counts are easy to distinguish.

The authors have addressed the issue of inflated plddts in these comments but then should describe this explicitly also in the manuscript.

b) The false-positive region could also be under-represented due to the choice of only sampling 1000 sequences under the arbitrary 1000 count threshold. How would this region change when sampling more low count sequences? How were the low count samples chosen? It would be good if the authors can provide insight into how the false positive versus true negative distribution changes with low-count sample size changes and changing the threshold.

This is a valid point as well. Due to computational limitations, we selected an equal arbitrary number of samples from each protein-minibinder complex. We still believe that the main point of being able to distinguish low- vs high counts is supported and will add an analysis of the FPs.

I maintain that it would be useful to analyse the FPs in more detail. A known true binder is picked up by the loss function, but the loss function cannot always distinguish FP from TP. As this loss function then forms the basis for the whole study, the authors need to do a better job of demonstrating its credibility. This matters especially downstream. E.g. later in the zero shot design section what is then the probability that the single design they get for up to 6.5% of targets are each likely to be false positives?

c) An AUC of 0.72 with 20% TPR at 10% FPR (Supp Fig 3) is not all that accurate, although does show utility of the loss function. The authors could comment on this also in light of exploring other loss functions.

This is true. We will make these limitations more clear. We note that this is highly dependent on the selected threshold as suggested above. We have added a point about this to the section regarding figure 2: *"We note that this is not a very good ratio, although it does increase the likelihood of successful binder selection"*.

Addressed.

6. Generating scaffold seeds from Foldseek is a great idea – and potentially very rich in initial scaffold information. How strong is the impact of the Foldseek database on binder design success? It could be instructive to show some statistics on the distribution of seeds (for a given set of targets) across different subsets of the Foldseek database used. The authors choose only the top ranked scaffold per target for their pipeline and subsequent analyses. But it would be helpful to assess the effects of integrating multiple seeds per target

into the pipeline and the downstream success. The best end binders may not necessarily derive from the top scaffold, and it would be good to get a quantitative handle on this.

This is hard to evaluate as we can only compare with known interfaces. We could sample more seeds for the same interface, but choose to evaluate only the one with the highest contact density to limit the computational cost and because this seems to be a determining factor of success rate (Figures 4d and 6d). In reality, many more seeds can be used and this is a parameter that can be set by the user. This is a question that has to be evaluated experimentally, however.

Agreed, and again good to mention that multiple to seeds could potentially be used to advantage in the actual manuscript.

7. A major drawback of the study is the over-focus on what the authors call the 'success rate', which is really a successful data-set coverage rate. The authors analyse whether their pipeline has significant coverage over the heteromeric data set, so they develop the study first with a zero-shot design approach (single sequence per target) and then with a convergence analysis to see if sampling more sequences per target allows them to increase the success/coverage of at least 1 successfully designed binder to more targets. This does not exceed 6.5% (185/2843) of the interfaces. That means their method couldn't find a single binder (<2Å RMSD) for most of this benchmark set. Whilst this kind of global-but-shallow coverage analysis provides some insights what is equally if not more meaningful is reporting how well their approach can reliably design multiple computationally 'successful' sequences for any given target (depth analysis) – with a view that these may be testable experimentally in the future to see how well predicted hits translate to real hits.

This may be interesting for some, but we believe less so than how many known interfaces can be covered. We thereby disagree with this remark and believe it is more meaningful to have binders towards many different interfaces than having many to one. This is because the main objective of binder design is to be able to design towards any interface. Most methods evaluate the depth, as suggested, and validate very few targets. This introduces a bias towards certain interfaces that are easier to bind and selected beforehand. Instead, we evaluate binding to all known non-redundant interfaces in the PDB. We believe this is the first study of its kind and this provides valuable insight into the severe shortcomings of current methods as only 6.5% of designs are successful according to computational evaluation. There are similarities with this across the field of protein structure prediction, notably that of protein docking where targeted evaluation sets were created and the reported performances were very good (for decades). Only when a much larger set was analysed could one see the inherent bias in these studies (see <https://www.nature.com/articles/s41467-022-28865-w>).

We believe that the greatest interest should lie in determining the probability of success of any binder among a fixed number of samples as this is what any potential user would be interested in (unless by coincidence we happen to evaluate exactly their target protein). We can add a report of how many successful designs are obtained for each successful target as well since this data is readily available, but think that this does not fulfil any meaningful purpose. We, therefore, refer to the data availability section for this metric. This same statement applies to the following as well:

I do not discount the validity of the coverage approach but the depth analysis is also important. Please see general comments above – also regards user interests. I also agree that having coverage as a metric is a good way to assess potential generality (and current lack thereof) and the authors could actually emphasise this in the manuscript.

In light of the above comments, I still maintain it would be very useful to report how many designs are obtained for each successful target.

For the binder convergence section, it would help to report the statistics on the distribution of successful binders across interface set – e.g. out of the 100 sequences sampled per target, how many sequences for each target were successful binders (<2Å RMSD)? The ‘depth’ success rate could be defined per target as the fraction of successful binder designs. The authors could then take a smaller subset of interfaces (ideally with several where they were unsuccessful) and push the depth of the method (exploring multiple seeds per target, much larger numbers of sequences per seed) to analyse the success rate per target as well as how many targets they can be successful at least once for. It is commendable that the authors include a section on analysing failed designs. Indeed, through this they come upon deeper insight into a potential correlate for improving coverage rate – namely, the dependence on contact density – this could additionally be integrated as a filter into their pipeline to enhance both coverage and depth – and actually lend confidence to their suggestion in the conclusion that: “This issue may be addressed by obtaining better scaffolds as we have shown that if scaffolds with high contact densities can be generated designs are more likely to be successful.”

The contact density is used as a parameter in the pipeline for selecting seeds and can be set by the user through the web interface:

https://colab.research.google.com/github/patrickbryant1/binder_design/blob/main/EvoBind.ipynb#scrollTo=twi_TCq9WUGY.

Addressed.

8. The study could benefit from analysis of a completely non-benchmark set. The authors could choose a small set of proteins not in any hetero or homomeric complex data set from which to perform a completely blind automated analysis. Then rather than choosing successful binders based on <2Å RMSD, they could reason the use of various thresholds in their scoring metrics to assign what are predicted successful binders. These could be tested by them or others in the future.

This is a great suggestion that we leave for a future study which includes experimental validation of new designs.

It’s important to mention this in the manuscript as a next logical step as the method would require this to demonstrate utility independent of pre-existing reference structures.

9. The conclusion section could benefit from a brief discussion on the following points:

- a) The authors don't report any experimental validation for their full binder design pipeline. It would be good if they could reason how well they predict their approach would be when tested experimentally.
- b) Does their approach apply to binder design for proteins in a specified conformation? Even though FoldSeek may work as before, AlphaFold might not be able to predict the target protein in the desired conformation which could maybe impede scoring.
- c) How do the authors envision improving the approach to get better scaffolds or improve the coverage and depth rate?

We thank you for these very wise suggestions. We have added these points to the conclusions section. We did not analyse the impact of AlphaFold's ability to sample the correct conformations. In general, AlphaFold has been found to prefer certain conformations as being most represented in the PDB and these are also the ones that are used for the seeds. It is possible that the wrong conformations are predicted and we mention this in the conclusions:

"We do expect that if a scaffold with a high contact density and low loss is obtained, accurate designs can be generated with EvoBind and these can be selected with high confidence. A factor that may impede the design is the possibility of predicting unintended conformations, which depends on the ability of AlphaFold2. As more structures are being solved at an accelerated pace, we do expect that better scaffolds will be available in the future, including for multiple conformations. "

Addressed.

Minor points:

1. Check use of italics and subscripts for equation, terms and in-text description

Still not addressed – pg 4 penultimate line 'di'

2. Correct Figure referencing – see Binder design convergence section: Figure 1 a -> Figure 5 a

Addressed.

3. The loss function is introduced in the context of comparison with native peptide binders. In this case the delta COM is with respect to the native peptide. It is not explained how the delta COM is defined in the design pipeline – presumably with respect to the chosen Foldseek structure – otherwise how would this be functional beyond native peptide comparison? The authors should clarify this.

This is correct, we have made this clear in the text.

"The Δ COM is taken towards the seed structure and added here to ensure the designs target the desired area and not its mirror image. Without it, it is possible to obtain the same loss on the opposite side of a protein. "

Addressed.

4. The Viterbi selection doesn't really in my opinion add much to the story and could be moved to the Supp Information for better flow.

We agree and removed this section.

Addressed.

5. There is no mention of the ESM-IF1 methionine bias in the main results, only as a technical point in the methods, yet the authors describe this in the abstract. Is this really necessary? If not, the authors should remove this from the abstract to improve the flow or alternatively present data in the results section.

We agree and removed this from the abstract.

Addressed.

6. The authors state that in the results that one sequence is chosen per target from the set of 2926 heteromeric interfaces. Then results are presented for $n=13156$ in Figure 4. In the methods the authors state that: 'In total, 13216/13893 designs were generated for the zero-shot evaluation'. None of these numbers match. Similarly in the binder convergence section they state: "resulting in successful designs towards 185 out of 2843 interfaces" for seeds of length 10 but $n_{10}=2895$ interfaces is stated in Figure 4 and in the methods: 'and 282853/292600 for the convergence analysis (100 per unique interface) using the heteromeric set' – which implies they used all the 2926 interfaces for the convergence analysis. The authors should clarify this, perhaps with an explanatory table or figure of analyzed systems across the manuscript.

Thank you for noticing this. The numbers had not been updated in all parts of the manuscript. Seeds could not be generated towards all interfaces for all lengths, which is why the number decreases with seed length.

For the convergence, we used a length of 10 and did not obtain predictions for all which is why the numbers differ from the zero-shot design in Figure 4. We did try to use all interfaces in all cases, but due to computational reasons, this was not always successful due to some jobs running out of time/memory.

We have added a table in the supplementary material (Supplementary table 1) and refer to this throughout the Figures/text.

Addressed.

7. The Off-target section should be renamed 'Designed binder specificity' or something similar. Off-target effects are a much larger topic and the authors don't demonstrate that designed binders wouldn't have off-target effects in the whole host of biomolecules they could bind to. Rather they provide a specificity test within the context of comparing against a subset of unintended targets. Furthermore, the plddt is only qualitatively suggestive. Indeed, in Fig 1 C there seems to be a small population of high plddt with moderate mutations contrary to what the authors claim and indeed some of the "off-target" regions in Figure 5d

have high plddts. Therefore, it is not clear how much real insight this section actually provides – it would be surprising if the plddt wasn't higher for the on-target effects. The authors could move this to the Supp Information.

We have renamed this section accordingly, but keep this in the current section since we find the higher pIDDT to be an important finding. It would be surprising if the pIDDT wasn't higher for the intended designs as this would mean the process does not work. This provides support to the specificity of the designs.

Addressed.

8. The improvement over ProteinMPNN looks encouraging but it seems not to be a fair comparison as the authors extract backbone atoms for EvoBind but only Ca atoms for ProteinMPNN, thus more limited information. It is therefore not surprising that EvoBind does better. The authors could compare using backbone weights across all backbone atoms in ProteinMPNN for a fair comparison.

We use the same amount of information when available. ProteinMPNN doesn't have the option of using the backbone atoms, but only CA. The N and C backbone atom positions provide a negligible amount of information compared to that of the CA positions. If one knows the CA positions, it is not very difficult to infer the positions of the other backbone atoms. We did not modify ProteinMPNN for the task of binder design. This could be an alternative method for which we have no preference. What we have observed is that if only backbone information is provided, ESM-IF1 seems to prefer better for homomers. Another reason for not choosing ProteinMPNN is that it can't be evaluated properly since it has been trained on almost all heterodimers in the PDB. This is a severe shortcoming of this method.

Addressed.

9. The following statement in the Methods is quite arbitrary: 'Generally, any amino acid sequence with a maximum of 50 residues is considered to be a peptide. The binders designed here are within this range.' Why 50? Can the authors substantiate this? If not, better to just say they designed peptide binders within a max of 50 amino acids.

We agree that this is arbitrary. This is the general consensus to our knowledge. We admit that there is no formal definition for what a peptide is and this goes for many concepts of biology (e.g. structural homology). We have changed this as suggested.

Addressed.

Reviewer #3 (Remarks to the Author):

The authors made substantial efforts to address all the comments made by Reviewer 3. The detailed point-by-point response is provided for each review comment. The comments have been addressed reasonably well. The work itself is quite interesting. The results are substantial. The analysis is convincing. Overall, the work provides a valuable method for peptide binder design.

Reviewers' comments:

The previous responses to reviewer 2 are marked in red (not bold) and the new responses are in bold. The new responses are in plain text (black, not bold).

Reviewer #1 (Remarks to the Author):

The manuscript by Bryant & Elofsson introduces a protocol on top of AlphaFold for the design of peptide binders of a target. The subject is topical and timely, and generally, I believe this could be a nice contribution. The authors cover binder design issues quite extensively, including specificity, which is nice.

However, I have several remarks, some more about form than substance.

1) I did not find the manuscript so easy to read.

We would have appreciated constructive criticism instead of a simple comment on the fact that the manuscript is complicated.

This is a result of extensive additions brought forward by many different reviewers (now 5 in total) over several rounds of review which include many, in our view, pedantic remarks. To include all points brought up, unfortunately, results in an extensive manuscript that is not that easy to read. We have tried to make the manuscript as easy to follow as possible by e.g. introducing the results section and have tried to improve upon this by addressing the points brought up in this round as well.

1.1) The manuscript is sometimes rather technical and several sentences/sections could be reformulated to get simpler. It is for instance the case of the second paragraph of the zero shot binder design section. I am not sure a standard reader could understand what is exactly done and why. Even I am unsure of the definition of a binder in this part. Is it just a fragment in contact with the target optimized using ESM-IF1? Is there any experimental validation of the binding? Please clarify. The sentence "In total, 205 successful designs were created towards 137 unique target interfaces" might be overstated.

We have changed the name of this section to "binder design". We do outline on several occasions what we define as successful binders, which is a standard taken from other studies (which we cite). These are predictions with $\text{RMSD} < 2\text{\AA}$ to the native binders. This is outlined 9 times in the text, including the figures. We have also added in the introduction to the results section that:

"We consider binders to be successful when predicted and native peptides have an $\text{RMSD} \leq 2\text{\AA}$." and hope the reviewer will find this sufficient.

We did not perform any experiments and made no such claims. We do believe that outlining what success means here, now, 10 times in total should be sufficient for the average reader.

There has, recently, been experimental validation of the EvoBind loss function including an earlier version of EvoBind using MC search compared to ESM-IF1 used here. This experiment demonstrates the high utility of the loss function and that design is rather the problem of searching for diverse sequences that generate low losses: <https://www.nature.com/articles/s42256-023-00691-9>

We have added a section on “experimental validation of the loss function” which we hope will convince the reviewer that the loss function is indeed functional if now the analysis of available data did not do so.

1.2) At some places, additional information could be welcome. Among others: (i) all symbols of Equation 1 are not defined, (ii) I could not find in the manuscript the definition of the NGS counts, etc. I really believe that a slight rewriting of the manuscript could greatly improve the impact for the reader.

What symbols are not defined in Equation 1? We assume this refers to m and n which we are sure the readers will understand are the number of atom pairs in the receptor and binder. We have added these in the description as well:

“

The loss is calculated after structural alignment on the target receptor protein. The binder pIDDT is the average pIDDT from AF over the binder, d_i is the shortest distance between all atoms in the receptor target atoms and any atom in the binder (**m pairs in total), d_j is the shortest distance between all atoms in the binder and any atom in the receptor target atoms (**n pairs in total**), and ΔCOM is the $\text{C}\alpha$ centre of the mass-weighted distance between the native and predicted binders. The ΔCOM is taken towards the seed structure and added here to ensure the designs target the desired area and not its mirror image. Obtaining the same loss on the opposite side of a protein is possible without it.**

“

We have written clearly in the section for the NGS counts (selecting true binders from experimental data) that:

“These binders were evaluated by counting the number of times they were detected to interact on yeast cell surfaces by Fluorescence-activated Cell Sorting (FACS), so-called Next-Generation Sequencing (NGS counts).”

2) The interest of not having - in my understanding - any control on the region of the receptor targeted by the binders is not obvious - zero shot design section. Usually, one wishes to target one specific region, and have control on it. I suggest the authors could elaborate a bit on the pros and cons of such strategy. Also, could the fact of not controlling the interacting regions result in designs that could be easier to tackle?

There is control. The idea here is to generate seeds = target regions using known structures. The user can then select what region to target. After all, one can't know what region may be good to target if one has no data for this. We realise that the

design concepts outlined here are novel and, perhaps, difficult to understand. We have outlined these to the best of our ability and recommend that the reviewer has another look on the explanations provided.

We have outlined the expected pros and cons of using this strategy in the discussion section which we do not wish to extend further. The analyses performed here regarding protein-peptide structure prediction, loss function evaluation on several data sets (now including independent experimental validation), seed generation and length, evaluation of zero-shot design, depth (100 sequences), off-target effects, and comparisons with other methods we believe are extensive enough to warrant publication and strongly discourage any suggestions for further additions.

3) To me, the section about binder specificity is not really convincing. Off-target issues often result from interactions with similar proteins, whereas here, the authors analyze the binding to unrelated proteins.

This is true. This analysis is to provide an intuition for the specificity towards the target interface - or class of interface. We have clarified this and added a point about the dissimilarity:

*“Even though the randomly selected proteins are dissimilar to the target proteins, the analysis helps to assess how specific EvoBind is towards a **certain class of interfaces.**”*

We have also added to the discussion that:

*“We find that the designed sequences are specific toward **the classes** of receptor interfaces, represented by median pIDDT values of 72 vs 45 towards unintended targets.”*

4) Some context about the perspective of having automated peptide binder design could be welcome. Having probes able to bind the target in silico/in vitro is really far away from something usable in vivo.

True. This is only for in-vitro design, and many more issues will surely arise in vivo. Still, today it is very difficult to generate a binder for in vitro applications and this will still be useful for a variety of applications (e.g. diagnostics or research). We feel that this discussion is outside the scope of this study as we make no claims about in vivo function and do not perform any in vitro experiments either (although the loss function has been independently validated in vitro). This is a computational study on available data to provide a scope for what can be expected of the latest technology instead of selecting 1-5 targets with known binders and simply designing towards these as in other studies.

Reviewer #2 (Remarks to the Author):

I have written my comments in blue coloured text within the remarks threads of the reviewer comments pdf document attached.

1. The subsection 'Optimal protein-peptide structure prediction' could be re-written more clearly. The authors should clarify that the standard AF2 structure prediction models do not alone give very accurate RMSD predictions for protein-peptide interfaces - only 12 out of 96 binders are predicted at their true location - even after optimizing the number of recycles. Nonetheless, they are still able to distinguish the binders that AF2 predicts to bind using the plddt score from AF2 (and coupled with their IF distance) – thus they have a good handle on true positives albeit with many false negatives (which should also be commented on in the manuscript).

We will make the false-negatives notion more clear. We have added the following: "The pLDDT of the peptide results in the highest AUC (0.94), and combining the pLDDT with the receptor interface dist results in a slightly higher true positive rate (TPR) at low false positive rate (FPR) with no reduction in AUC, although many positives are missed (84 out of 96). "

It would be good to emphasise this in the following sentence too, e.g. "Together, this suggests that although AF is largely inaccurate at predicting protein-peptide interactions, when it does so accurately the peptide will be situated close to the target interface and have a high pLDDT score."

We have added this as suggested.

2. The authors could consider plotting the data from Supp Fig 1 by binning by mutation fraction, calculating the correlation between plddt and IF distance and plotting correlation vs mutation fraction bin. This would be insightful for the main manuscript as a new Fig 1 d. Perhaps the term binning caused confusion. I will clarify. For each value of mutation fraction, determine the correlation between IF and plddt for the corresponding vertical line of points. Then plot correlation against mutation fraction. If what the authors state is true correlation should decrease with mutation fraction and this would in my opinion be easier to interpret rather than multiple columns of dots.

The reviewer wants us to take each mutation fraction and correlate pLDDT and IF-distance. If this decreases with increasing fraction the reviewer claims that the correlation is true. Why is this? This seems to measure the noise within each bin which the reviewer claims, then, should increase with the mutation fraction. It could easily be that this is not true and we do not see the point with correlating between bins.

What is important is that the distance increases with the mutation fraction and pLDDT and not the exact noise level at each mutation fraction. We don't see how this would be easier to interpret for any reader either.

3. Why do the authors choose the exact loss function they do and how does it compare with other potential choices? Although rationale is provided for the plddt and the IF distances, it is

not for the COM – and indeed many combinations of terms are possible (and easily explorable once structures have been predicted). The authors could assess several additional loss functions and rank them. Why did the authors choose AF2 model_1 instead of the further fine-tuned model_1_ptm, which would provide the option of using the predicted aligned error (pAE) as a way of scoring binder candidates and that could potentially be integrated into more effective loss functions? Even though not explicitly computed in this study, the concept of using other loss functions, their potential advantages/disadvantages computational costs etc. should be discussed more thoroughly in the conclusions

In model_1, no predicted TM-score (pTM) or predicted aligned error (pAE) is available, only the predicted IDDT (pIDDT) for each residue. We state this in the methods section. The reason for using model_1 is because this worked the best for predicting heterodimeric complexes according to our previous studies: <https://www.nature.com/articles/s41467-022-28865-w/tables/1>.

We added a note about this in the methods section:

We used model_1 since this has been found to be optimal for predicting heterodimeric complexes, which is a highly related task (<https://www.nature.com/articles/s41467-022-28865-w/tables/1>).

One could of course explore all possible combinations of loss functions, but we do not feel that this is important as long as a good enough loss function exists. There are limitations to all studies and urge the reviewer to recognise this and that all possible additions can't be discussed at length. We have shown through independent evaluations (now also experimental from others) that our loss function is accurate. We leave the development of the loss function further to future studies.

4. The validation of the loss function against the experimental miniprotein binder data set is an elegant idea. A few points should be considered here:

a) It would help to plot the 1000 NGS count threshold here on Figure 2. Although Figure 2 does show a strong transition from high to low loss on an increase in NGS counts, there seems still to be a significant false-positive (lower-left corner) distribution. Could this be explained by the reports that AlphaFold has been shown to predict inflated pLDDTs for short spurious proteins (see e.g. Monzon, Haft & Bateman 2022 doi.org/10.1093/bioadv/vbab043 for results on AntiFam). If so, do the authors have a strategy to potentially mitigate this issue?

This is a good point and we have analysed the FPR along with the inflated pLDDTs by taking the examples with normalised NGS counts < 0.01 and a loss < 1. These have a median plddt of 85.0, which is slightly higher than the median of all examples with normalised NGS counts < 0.01 (83.6). Note that all pLDDTs are quite high in this case, which is a characteristic of sequences designed with classical methods such as Rosetta - they are more stable and therefore easier to predict in single sequence mode.

We have added to the section regarding Figure 2: "We note that this is not a very good ratio, although it does increase the likelihood of successful binder selection".

The NGS count threshold was set to 0.01 after normalising with the highest count for each protein-binder pair. The 1000 NGS counts refer to the sampling (1000 sequences below 1000 NGS counts) used to reduce the computational requirements as there were 172581 sequences available from this study in total. Plotting the 0.01 selection threshold in Figure 2 we do not think is necessary as the densities at high/low loss/NGS counts are easy to distinguish.

The authors have addressed the issue of inflated plddts in these comments but then should describe this explicitly also in the manuscript.

We have added a supplementary note about this as we see this as more of a side track than being part of the main analysis:

“Regarding the analysis of the designed minibinders in Figure 2, we have analysed the false positive rate (FPR) along with the inflated pIDDTs (compared to the peptide binders in Figure 1) by taking the examples with normalised NGS counts<0.01 and a loss <1. These have a median plddt of 85.0, which is slightly higher than the median of all examples with normalised NGS counts<0.01 (83.6). Note that all pIDDTs are quite high in this case, which is a characteristic of sequences designed with classical methods such as Rosetta. Designed proteins are more stable and therefore easier to predict in single sequence mode as can be seen for e.g. Top7 here: [biorxiv.org/content/10.1101/2021.08.24.457549v1.full.pdf](https://www.biorxiv.org/content/10.1101/2021.08.24.457549v1.full.pdf) and here: <https://www.biorxiv.org/content/10.1101/2022.12.13.520346v1.full.pdf>. “

5. The false-positive region could also be under-represented due to the choice of only sampling 1000 sequences under the arbitrary 1000 count threshold. How would this region change when sampling more low count sequences? How were the low count samples chosen? It would be good if the authors can provide insight into how the false positive versus true negative distribution changes with low-count sample size changes and changing the threshold.

This is a valid point as well. Due to computational limitations, we selected an equal arbitrary number of samples from each protein-minibinder complex. We still believe that the main point of being able to distinguish low- vs high counts is supported and will add an analysis of the FPs.

I maintain that it would be useful to analyse the FPs in more detail. A known true binder is picked up by the loss function, but the loss function cannot always distinguish FP from TP. As this loss function then forms the basis for the whole study, the authors need to do a better job of demonstrating its credibility. This matters especially downstream. E.g. later in the zero shot design section what is then the probability that the single design they get for up to 6.5% of targets are each likely to be false positives?

It is true that the FPs matter throughout the paper. The FP rate is expected to be as estimated in the initial analysis of the 12 peptide binders. This means that if 6.5% of designs are

“successful” according to the loss function, a fraction as specified by the ROC-curve in Figure 1 will be FPs. As we state “At an FPR of 10%, 87% of successful designs ($\text{RMSD} \leq 2 \text{ \AA}$) can be selected.”. Translating this to the example provided by the reviewer this means that 0.65% will be FPs according to the same selection.

We do not see how the reviewer aims for this to be analysed in further detail. We have demonstrated the credibility of the loss function on independent test sets as specified in Figures 1 and 2. Further experimental analyses are necessary to truly assess the validity of the loss function which we have stated several times to be outside the scope of the study here which is purely computational.

In addition, an independent laboratory test of the developed loss function has been performed (now published in Nature, called EvoPlay: <https://www.nature.com/articles/s42256-023-00691-9>). If the reviewer has objections to the computational assessment here we suggest to also considering the strong selection bias in similar experimental studies and raising further concerns with the editor as our approach here has been accepted in the transfer from Nature Communications.

6. Generating scaffold seeds from Foldseek is a great idea – and potentially very rich in initial scaffold information. How strong is the impact of the Foldseek database on binder design success? It could be instructive to show some statistics on the distribution of seeds (for a given set of targets) across different subsets of the Foldseek database used. The authors choose only the top ranked scaffold per target for their pipeline and subsequent analyses. But it would be helpful to assess the effects of integrating multiple seeds per target into the pipeline and the downstream success. The best end binders may not necessarily derive from the top scaffold, and it would be good to get a quantitative handle on this.

This is hard to evaluate as we can only compare with known interfaces. We could sample more seeds for the same interface, but choose to evaluate only the one with the highest contact density to limit the computational cost and because this seems to be a determining factor of success rate (Figures 4d and 6d). In reality, many more seeds can be used and this is a parameter that can be set by the user. This is a question that has to be evaluated experimentally, however.

Agreed, and again good to mention that multiple to seeds could potentially be used to advantage in the actual manuscript.

We have added this here:

...complexes corresponding to all (2926) unique heteromeric protein interfaces in the PDB. One seed per target is evaluated here, although many more seeds can be used. Importantly, none of the methods in EvoBind has seen any of these complexes before, as they are trained only on single-chain proteins.

7. I still maintain it would be very useful to report how many designs are obtained for each successful target.

We have not added this. We feel that there are already too many analyses in this manuscript and the interested reader can perform this analysis easily themselves from the provided data (simply select on RMSD<2).

8. The study could benefit from analysis of a completely non-benchmark set. The authors could choose a small set of proteins not in any hetero or homomeric complex data set from which to perform a completely blind automated analysis. Then rather than choosing successful binders based on <2Å RMSD, they could reason the use of various thresholds in their scoring metrics to assign what are predicted successful binders. These could be tested by them or others in the future. It's important to mention this in the manuscript as a next logical step as the method would require this to demonstrate utility independent of pre-existing reference structures.

We leave this to future studies and refrain from any further additions to the manuscript. These should have been brought up in the first round of review if the reviewer considered these to be essential.

9. Check use of italics and subscripts for equation, terms and in-text description
Still not addressed – pg 4 penultimate line 'di'

We have addressed these points.

4.

Reviewer #3 (Remarks to the Author):

The authors made substantial efforts to address all the comments made by Reviewer 3. The detailed point-by-point response is provided for each review comment. The comments have been addressed reasonably well. The work itself is quite interesting. The results are substantial. The analysis is convincing. Overall, the work provides a valuable method for peptide binder design.

We thank reviewer 3 for acknowledging the effort that has gone into generating the analyses presented here and for the appreciation of the results themselves which we are sure will become more apparent as more independent laboratory tests are published.

REVIEWERS' COMMENTS:

Reviewer #1 (Remarks to the Author):

Most of the comments about the content were handled quite well, even if sometimes in a minimalist way. For example, the point on the specificity of the designed binders (ref. 1, remark 3) could have benefited from an analysis, for certain classes of interfaces, of the distribution of scores within the class. This seemed a marginal effort compared to the large amount of work done, to feel how the procedure is likely to behave for identifying an off-target binder. This does not diminish the impact of the contribution, but adding a sentence of perspective on this could probably be welcomed.